# Oxygen Gas Sensing Using a Hydrogel-Based Organic Electrochemical Transistor for Work Safety Applications

**DOI:** 10.3390/polym14051022

**Published:** 2022-03-03

**Authors:** Francesco Decataldo, Filippo Bonafè, Federica Mariani, Martina Serafini, Marta Tessarolo, Isacco Gualandi, Erika Scavetta, Beatrice Fraboni

**Affiliations:** 1Department of Physics and Astronomy, Alma Mater Studiorum—University of Bologna, Viale Berti Pichat 6/2, 40127 Bologna, Italy; filippo.bonafe4@unibo.it (F.B.); marta.tessarolo3@unibo.it (M.T.); beatrice.fraboni@unibo.it (B.F.); 2Department of Industrial Chemistry, Alma Mater Studiorum—University of Bologna, Viale Risorgimento 4, 40136 Bologna, Italy; federica.mariani8@unibo.it (F.M.); martina.serafini6@unibo.it (M.S.); isacco.gualandi2@unibo.it (I.G.); erika.scavetta2@unibo.it (E.S.)

**Keywords:** oxygen sensor, PEDOT:PSS, organic electrochemical transistor, hydrogel, work safety

## Abstract

Oxygen depletion in confined spaces represents one of the most serious and underestimated dangers for workers. Despite the existence of several commercially available and widely used gas oxygen sensors, injuries and deaths from reduced oxygen levels are still more common than for other hazardous gases. Here, we present hydrogel-based organic electrochemical transistors (OECTs) made with the conducting polymer poly(3,4-ethylenedioxythiophene): poly(styrene sulfonate) (PEDOT:PSS) as wearable and real-time oxygen gas sensors. After comparing OECT performances using liquid and hydrogel electrolytes, we identified the best PEDOT:PSS active layer and hydrogel coating (30 µm) combination for sensing oxygen in the concentration range of 13–21% (*v*/*v*), critical for work safety applications. The fast O_2_ solubilization in the hydrogel allowed for gaseous oxygen transduction in an electrical signal thanks to the electrocatalytic activity of PEDOT:PSS, while OECT architecture amplified the response (gain ~ 104). OECTs proved to have comparable sensitivities if fabricated on glass and thin plastic substrates, (−12.2 ± 0.6) and (−15.4 ± 0.4) µA/dec, respectively, with low power consumption (<40 µW). Sample bending does not influence the device response, demonstrating that our real-time conformable and lightweight sensor could be implemented as a wearable, noninvasive safety tool for operators working in potentially hazardous confined spaces.

## 1. Introduction

Except for small living entities still relying on less efficient, anaerobic metabolisms, most animals depend on oxygen, as it plays an important role in their energy-generating processes [1]. Human beings metabolize around 200 g of oxygen daily and could not live without an O_2_ supply for longer than 10 min, with potentially irreversible neural damage (neuron death) already occurring after few minutes [2]. Its importance is further highlighted by the fact that, despite the existence of different hazardous gases known for their toxicity and lethal effects, oxygen depletion is still one of the major causes of death and injuries and is often underestimated [3]. Especially in confined spaces, oxygen-concentration reductions can be caused by leakage of inert gases, such as nitrogen, helium, or argon that are often used in laboratories, medical clinics or industrial facilities. In addition, stored ripening fruits, organic material decomposition, combustion, bacteria, and rusting metals could contribute in oxygen consumption, thus producing an oxygen-depleted environment [4,5]. In the last decade, reports by the Centers for Disease Control and Prevention (CDC) and the U.S. Bureau of Labor Statistics demonstrated that oxygen deficiency in confined spaces is a serious issue, causing a number of deaths each year [6,7]. It is worth noting that the Occupational Safety and Health Administration (OSHA) highlights the hazards and risks of confined spaces, requiring gas monitors to prevent work injuries.

Nowadays, robust and reliable oxygen sensors are commercially available, either continuously monitoring the room gas percentage or working as portable systems for the single end-user [8,9,10,11,12]. However, these technologies are expensive and cannot be included as a standard equipment for each worker. Thus, wearable oxygen sensors directly integrated on worker overalls or helmets as personal protective equipment (PPE) could mitigate this issue, providing continuous monitoring of the gaseous oxygen level in the wearer’s surroundings. 

Wearable sensors need to be biocompatible, flexible, lightweight and conformable to be non-invasive and easily worn. Nowadays, there is heightened interest in wearable technologies, as highlighted by several reviews, reporting up-to-date advances and implementation of chemical, gas and pressure/strain sensors aiming at meeting the above mentioned needs [13,14,15,16]. Among these sensors, conducting polymer-based devices have been recently investigated due to their flexibility, stretchability and low-cost which allow the production of comfortable devices suitable for daily use [17,18,19]. 

In our previous work, we proposed poly(3,4-ethylenedioxythiophene):poly(styrene sulfonate) (PEDOT:PSS) organic electrochemical transistors (OECTs) to monitor dissolved oxygen concentration in cell mediums for biological applications. [20] OECTs are three-terminal devices in which the electronic current flowing in the semiconducting channel is modulated by means of an electrolyte solution throughout the application of a low potential (<1 V) on the gate. Gate voltage allows for ion injection/extraction in/from the polymeric matrix, reversibly changing its redox state, thus its electrical conductivity. OECTs have been widely used as chemical sensing devices (analytes, biomarkers, molecules) [21,22,23,24,25,26,27,28] since they allow for low power consumption, ease of functionalization and inherent signal amplification and filtering. In particular, our devices exploited PEDOT:PSS electrocatalytic activity toward oxygen when it was negatively polarized with respect to a reference electrode. Consequently, the occurring oxygen reduction reaction led to polaronic state formation without the need of external dopants or specific functionalization, as previously modelled by Singh et al. using density functional theory [29]. The energetically preferable reaction is the following:(1)$$4{\mathrm{Pedot}}^{0}+O_{2}+4H^{+}\to 4{\mathrm{Pedot}}^{+}+2H_{2}O$$
which causes the injection of holes in partially oxidized Pedot:Pss. When Vg and Vd are set to reach the conditions that allow the electrocatalysis at the PEDOT:PSS channel (positive gate bias triggering device turn off), oxygen reduction takes place, enhancing the conductivity and the current that flows in the conducting polymer channel. 

In the present work, working to develop a reliable and low-cost gas oxygen sensor, we moved from the detection of dissolved oxygen in solution to the identification of gaseous oxygen in a confined space in the target range for work-safety applications, 13–21% O_2_, without any specific functionalization of the sensing PEDOT:PSS film. Since the electrolyte solution, which is an important component in an OECT architecture, does not reach a rapid equilibrium with the surrounding environment, we substituted the liquid electrolyte with a conducting agarose-based hydrogel coating. In this way, we enabled electrochemical sensing on gaseous compounds permeating/dissolving in the hydrogel, foreseeing a compact and real-time wearable sensor. The transistor architecture provides a high output signal amplification (10^3^–10^4^) owing also to the extremely low gate current (leakage current).

Wearable sensor devices were patterned on thin plastic substrates, proving the repeatability, reproducibility and preservation of the performances under bending.

The here proposed low-cost wearable OECTs patterned on plastic provide proof-of-concept sensors for real-time and reliable gas-oxygen monitoring with low power consumption (30–40 µW). The device integration on worker overalls/helmets would help keep the wearer safe constantly, alerting the user of oxygen depletion in confined spaces with low invasiveness. 

## 2. Materials and Methods

### 2.1. OECT Fabrication

Glass/plastic substrates were cleaned by sonication in acetone/isopropanol/distilled water baths. Afterward, substrates were dehydrated for 10 min at 110 °C and Microposit S1818 positive photoresist was spin coated (4000 rpm for 60 s) and annealed at 110 °C for 1 min. Metallic contacts were patterned through direct laser lithography by using the ML3 Microwriter (from Durham Magneto Optics, Cambridge, UK). The photoresist was developed with a Microposit MF-319 developer. Then, 10 nm of chromium and 30 nm of gold were deposited by thermal evaporation. Samples were immersed in acetone for 4 h for photoresist liftoff and then rinsed by sonication in acetone/isopropanol/distilled water baths. A double layer of S1818 was deposited for the photolithography of the PEDOT:PSS channel [30]. After the development, substrates were treated with air plasma (15 W for 4 min) and the PEDOT:PSS solution was spin coated at 3000 rpm for 10 s. The solution was made of 93.75% PEDOT:PSS (Clevios PH1000, provided by Heraeus Deutschland GmbH & Co., Leverkusen, Germany) with 5% ethylene glycol (EG) (Sigma Aldrich, St. Louis, MO, USA), 1% 3-glycidoxypropyltrimethoxysilane (GOPS) (Sigma Aldrich, St. Louis, MO, USA) and 0.25% 4-dodecylbenzenesulfonicacid (DBSA) (Sigma Aldrich, St. Louis, MO, USA). This suspension was treated in an ultrasonic bath for 15 min and filtered using 1.2 μm cellulose acetate filters (Sartorius) before the deposition. The resulting film thickness was (100 ± 10) nm. The samples were subsequently baked at 120 °C for 1 h. Then, the photoresist was lifted off following the procedure reported above, and devices were submerged in distilled H_2_O for 1 h and dried with a nitrogen flux. The resulting OECTs presented both channel and gate in PEDOT:PSS, the first having length (L) and width (W) of 1 mm and 0.3 mm, respectively, while the latter dimensions being 1.9 mm × 0.8 mm. Channel to gate distance was d = 1.6 mm.

A phosphate buffer solution was prepared dissolving 0.1 M of H_2_KPO_4_ in water and adjusting the pH with a concentrated solution of NaOH, reaching a final value of 5.5. Hydrogel formulation was prepared dissolving 0.7% of agarose in 0.1 M PBS (described above) at 90 °C, obtaining a viscous but clear solution. Once removed from the hotplate, the hydrogel was then ready to use. In particular, devices with the thin hydrogel (30 µm) layer were further refined by dip-coating the sample in the hydrogel formulation for 10 s and drying it at room temperature for 10 min. OECTs having the 0.1 M PBS or the thick hydrogel (5 mm) as gating electrolyte were completed using a transparent, cylindrical polydimethylsiloxane (PDMS) well to contain the electrolyte. Devices having the thin or thick hydrogel coating are referred to as H-Thin and H-Thick throughout this work. H-Thin OECTs were finally immersed in glycerol for 10 s and left at room temperature for 1 h to limit water loss over time. Similarly, H-Thin OECTs were fabricated onto transparent, 125 µm thick polyethylene napthalate (PEN) films to yield sensors on flexible substrates.

### 2.2. Electrical Characterization

Electrical data for the transcharacteristic (transfer), characteristic (output) and Id(t) curves were acquired using a Keysight B2912A (Keysight Technologies, Santa Rosa, CA, USA) for channel and gate biasing. Transfer curves were obtained with a constant bias of −0.3 V on the channel (Vd), sweeping the gate potential (Vg) between −0.2 V and 0.7 V with a scan rate of 0.04 V/s. The fifth cycle (stable response) was taken for plotting both the transfer and transconductance (first order derivative of the transfer) curves. Output curves were studied sweeping the Vd between 0 and −0.4 V with a scan rate of 0.04 V/s, for the eight different constant Vg (from 0 V to 0.7 V, step = 0.1 V). Id(t) plots were studied with constant polarizations of the channel, −0.3 V, and on the gate, 0.3 V, the last one selected for maximum OECT transconductance. 

Pulsed experiments for analyzing the dynamic behavior of the OECTs were carried out with constant Vd = −0.1 V, and a square wave potential on the gate electrode, from V_g(OFF)_ = 0 V to V_g(ON)_ = 0.3 V, with 25% of duty cycle. The cycle duration lasted 0.4 s for the devices having 0.1 M PBS and the thick hydrogel as electrolyte, and 20 s for the thin hydrogel-based device.

AC measurements were performed with the MFLI lock-in amplifier (from Zurich Instruments, Zurich, Switzerland). The OECTs were biased with constant DC drain and gate voltages corresponding to their maximum transconductance (V_g_ = 0.3 V and V_d_ = −0.3 V, respectively). A small sinusoidal oscillation with amplitude |v_g,AC_| = 100 mV and angular frequency of ω was then applied to the gate electrode. This led to an AC current in the transistor channel, whose amplitude |i_d,AC_| was measured from the drain terminal with the lock-in amplifier. During each acquisition, the modulation frequency was swept between 0.1 and 103 Hz to acquire the current spectrum of the sensor. The OECT transconductance was then calculated as a function of the frequency assuming a linear response of the device (g_m,AC_ = |i_d,AC_|/|v_g,AC_|). 

### 2.3. Data Analysis

The device dynamic time response (τ) was extracted using the following exponential decay function [31,32]: (2)$$I_{d}=I_{d,o}\times {\mathrm{Ae}}^{-\frac{\left(t-t_{0}\right)}{\tau }}$$

The sensitivity of device sensing O_2_ was extracted as the slope of the linear fit in the range under study. The calibration line was obtained linearly fitting Id versus the logarithm of the O_2_ percentage.

## 3. Results

### 3.1. PBS-vs.-Hydrogel as Gating Electrolyte for OECTs

Organic electrochemical transistors (OECTs), having 1 mm × 0.3 mm (L × W) and 1.9 mm × 0.8 mm PEDOT:PSS channel and gate respectively, were fabricated using direct writing lithography, as outlined in Materials and Methods. PEDOT:PSS was chosen as the active material because it allows for oxygen detection and quantification without any particular functionalization, as demonstrated in our work on dissolved-oxygen sensing using PEDOT:PSS-based OECTs. Similarly, the device dimensions were scaled down, maintaining the same ratio between the gate and channel active areas of the previous work, A_g_/A_ch_ = 5. Indeed, the larger gate area allowed for a greater potential drop on the channel/electrolyte interface, where the oxygen reaction takes place [20]. In order to move from sensing the oxygen dissolved in liquid to the study of the oxygen in gas phase, we combined the OECT with a phosphate buffer saline (PBS)-based agarose hydrogel, which supplied the ions for the PEDOT:PSS channel doping/dedoping. We then compared the device performances gated using liquid PBS or the hydrogel, studying two different thicknesses, namely an extremely thin coating (H-Thin, 30 µm) and a very thick one (H-Thick, 5 mm). H-Thin, PBS and H-Thick OECT schematic renderings are reported in Figure 1a–c, respectively. For the experiments involving PBS and the thick hydrogel, a transparent, cylindrical Polydimethylsiloxane (PDMS) well was used to contain the electrolyte, as shown in Figure 1b. The complete device characterization is reported in Figure 1 and Appendix A. It is worth noting that OECTs display current modulation also when the standard electrolyte PBS is substituted by the hydrogel as an electrolyte for gating, as shown in Figure 1d and Appendix A for transcharacteristic and output curves, respectively. Conversely, a slight reduction (around 30%) in the transconductance (g_m_) peak is reported in Figure 1e for the hydrogel-based OECTs, together with a slight voltage shift for the H-Thin one. Despite similar performances in the steady-state behavior, a significant difference in the device dynamic response and in the frequency domain was evident for the thin hydrogel-based OECT. Figure 1f compares the average time response (τ) upon a positive, square voltage polarization on the gate (switching between the “on” and “off” state) of four OECTs per type. H-Thin OECT τ values underwent an increase in the time response of three orders of magnitude (as visible also in the time scales of the raw drain currents in Appendix A). We ascribed this increment to the enhancement of the ionic resistance, caused by the thinner hydrogel layer, as suggested by the reduction of its gate current, I_g_ (Appendix A). PBS-based and H-Thick samples had similar time responses and Ig values. The transconductance versus frequency analysis confirmed the slower response of the H-Thin OECT, highlighted by the shift of the curve towards smaller frequencies (Figure 1g). However, it is worth noting that the H-Thin OECT time response can still be considered low (<0.5 s) and that the lower gate current leads to an increased gain of the device, compared to PBS-based and H-Thick OECTs. A summary of the described device parameters is outlined in Table 1, assessing the effective OECT gating using agarose hydrogel as an electrolyte.

### 3.2. Oxygen Gas Sensing Using OECTs on Glass Substrates

Oxygen gas sensing tests were carried out fluxing gas streams containing O_2_/N_2_ at different concentrations (starting from a nitrogen-saturated environment to set the blank baseline of the device). The schematic setup is reported in Appendix A. It is worth noting that the H-Thick OECT did not show significant responses to the four increasing oxygen additions (starting from a nitrogen-saturated environment) in the range applicable for work safety (Appendix A), probably owing to the thickness of the hydrogel layer whose oxygen permeability is not fast enough for sensing in the selected time scale. Only a slight drift of the current from the baseline was observed, which cannot be correlated with O_2_ additions. Since we aimed for a device capable of quickly warning potential oxygen deficiencies in the workplace, the thicker hydrogel presented unsuitable performances for our purposes. In contrast, a fast response to O_2_ variations was obtained with the thin hydrogel coating. H-Thin OECTs were thus chosen for oxygen sensing measurements and were further improved by adding a glycerol treatment, which, as highlighted in our previous work, improved the stability of the current output by limiting water loss in the hydrogel [26]. Appendix A proves that glycerol does not alter the electrical performances of the OECTs.

Figure 2a reports the output current increase of a H-Thin OECT upon several oxygen additions (starting from a nitrogen saturated environment) in the sealed chamber, with V_d_ = −0.3 V and V_g_ = 0.3 V, chosen for maximum transconductance. We hypothesize that, owing to the thin hydrogel layer, O_2_ rapidly dissolves in the electrolyte and electrochemically interacts as oxidant agent with the PEDOT:PSS channel, doping the semiconducting material, and increasing its conductivity. The energetically favorable reaction (no need of a catalyst or external dopants to ease electrocatalysis), proposed by Singh et al. using the density functional theory, ref. [29] is reported in Equation (1).

OECT output current values were linearly correlated to the logarithm of the O_2_ concentration fluxed in the chamber, as demonstrated in Figure 2b, thus allowing quantitative oxygen sensing with very low power consumption (<40 µW). Three sensors, fabricated in different batches, reported comparable sensitivities (see Table 2), confirming the reproducibility of the devices and allowing for the extraction of an average sensitivity of (−12.1 ± 0.6) µA/dec.

### 3.3. Oxygen Gas Sensing Using OECTs on Flexible PEN Substrates

To obtain a wearable device that could be integrated into a worker’s helmet or overalls, we fabricated the sensor on a thin plastic substrate of polyethylene napthalate (PEN), having a thickness of 125 µm. Replicating the O_2_ sensing test and device calibration (Figure 3a,b, respectively), we assessed similar sensitivities for the H-Thin OECT on plastic and on glass. Again, the reproducibility of three devices was studied (Table 3), obtaining comparable results, with an average sensitivity of (−15.4 ± 0.5) µA/dec, slightly higher than the one calculated for OECTs on glass. It is worth noting that OECT sensing using the drain current allows for a gain factor on the sensitivities higher than 104 and reduces the noise incoming from the gate current, since the slow time-responding transistor acts as a low pass filter, as explained in our previous work [20]. Appendix A highlights the gate current variation upon the oxygen addition of OECT n°2 and the extracted calibration curves. A software smoothing process was required to reduce the noisy gate signal, which however maintains fluctuations and spikes especially during or close to the O_2_ additions. Furthermore, the extracted sensitivity was below nA/dec. Comparing Appendix A with Figure 3 highlights the importance of the filtering and amplification provided by the OECT architecture.

Aiming at a robust and reliable sensor, in addition to reproducibility, we also measured the device repeatability and its responses to oxygen additions or removals in the “work-safety” range of 13–20%. Figure 4a highlights the stability of the device response for two consecutive measurements, with I_0_ representing the current flowing into the channel of the device when no oxygen was present (nitrogen-saturated environment), which was subtracted to exclude the baseline shift (slight reduction) upon sensor reuse. The graph shows almost superimposed I_d_-I_0_ values for the oxygen additions with sensitivities of (−13.2 ± 1.6) µA/dec and (−14.9 ± 0.7) µA/dec, that are comparable within their standard deviations. The device current response to oxygen additions and removals is reported in Figure 4b, with the analysis displayed in Figure 4c. Again, comparable sensitivities are obtained, highlighting the sensor’s robust response that does not depend on the previous O_2_ concentrations.

Bending tests and real-time warning tests were carried out to evaluate sensing performances while mimicking real-life applications using a portable electronic readout wirelessly connected to a smartphone application for real-time data acquisition. We chose a bending radius of 6 mm (Figure 5a), simulating a device integrated onto the work suit or bracelet of a worker. Figure 5b shows that a similar sensor response upon oxygen additions was obtained for the flat and, subsequently, bent sample: the curve slopes are identical, as shown by the ratio between the extracted sensitivity, 1.03 ± 0.02 a.u. We hypothesized that the current reduction was caused by sample manipulation to obtain the bending configuration in Figure 5a, since repeated flat/bent tests showed independent current from the sample bending (Appendix A). The portable electronic reader is shown in Figure 5c, powered by a coin battery allowing drain (V_d_ = −0.3 V) and gate (V_g_ = 0.3 V) biasing of the sensor. Upon alternating low oxygen concentrations with the standard oxygen levels present in breathable air in the sealed chamber, we assessed (Figure 5d) that the current monitored by the portable system follows oxygen contents, proving that our device could be employed as wearable, battery powered, low-cost O_2_ sensor. Finally, it is noteworthy that once calibrated, a safety threshold current may be selected and set to activate a real-time alert value to protect the wearer in case of severe oxygen depletion before oxygen deficiencies cause breathing fatigue or impaired judgment.

## 4. Discussion

Oxygen depletion in confined spaces caused several injuries and deaths in the last decade. OSHA considers O_2_ levels below the threshold of 19.5% in air as oxygen deficiency, since even small reductions from the optimal value of 20.9% may cause dizziness and fatigue, impairing people from escaping or calling for help.

Our work focused on the design and realization of a real-time, wearable and lightweight oxygen sensor, continuously monitoring the oxygen percentage in the range [13–21%] for operators’ safety. We employed a thin hydrogel as a semi-solid electrolyte for gating a PEDOT:PSS OECT (H-Thin OECT), patterned on a PEN plastic substrate. The ~30 µm thin PBS-based hydrogel coating allowed for an effective quantitative O_2_ sensing and granted ease-of-use to device handling and wearability, since the small microfabricated device could noninvasively be added to worker overalls or helmets as standard personal protective equipment. The repeatability and reproducibility assessed the reliability of the sensor, whose current was linearly correlated to the logarithm of the oxygen percentage (sensitivity = −12.2 ± 0.6 µA/dec). Bending tests demonstrated that the proposed sensors could be easily conformed to a curved surface or a garment without performance loss. Finally, we demonstrated that the sensors could easily interface with a portable reader for real-time sensing in wearable applications, allowing for selection of an O_2_ threatening threshold for safety. The proposed low-cost wearable OECTs patterned on plastic provide proof-of-concept sensors for real-time and reliable gas-oxygen monitoring with low power consumption (30–40 µW). Different from commercially available portable O_2_ sensing devices, it can be directly integrated on workers’ overalls or helmets and wirelessly connected to a user’s smartphone. This paves the way toward the achievement of the “wear-and-forget” functionality, noninvasively but constantly protecting the wearer from potential oxygen deficiencies. 

## Figures and Tables

**Figure 1 polymers-14-01022-f001:**
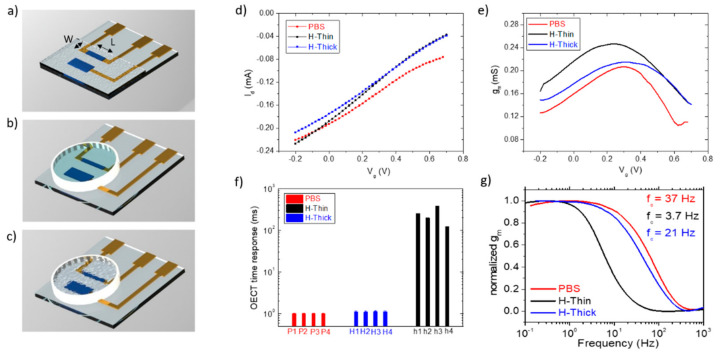
OECT configuration and electrical characterization. Rendering of the OECT devices on glass used with the thin hydrogel coating (**a**) and the thick hydrogel (**b**) or PBS electrolytes (**c**); (**d**) transfer; (**e**) transconductance; (**f**) time response to a voltage pulse on the gate of the OECT having PBS (red); the thin hydrogel (blue), or the thick one (black), as an electrolyte for gating. (**g**) Transconductance versus frequency measurements for OECTs having PBS (red); the thin hydrogel (blue); or the thick one (black); as electrolyte for gating, with relative cutoff frequencies (corresponding to the—3 dB points) extracted at 37, 3.7 and 21 Hz, respectively.

**Figure 2 polymers-14-01022-f002:**
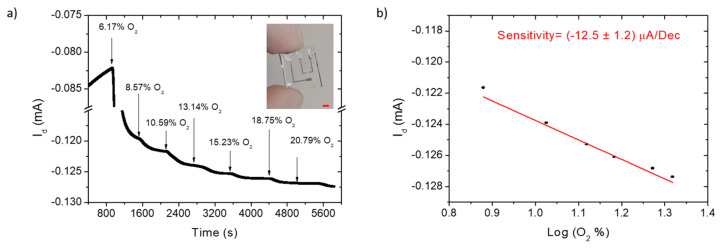
H-Thin OECT on glass substrates for oxygen sensing: I_d_(t) measurement (**a**); device calibration (**b**); for oxygen addition (black arrows) in the sealed cell, starting from a nitrogen-saturated environment (0% of O_2_); device picture is reported in the inset of (**a**); with a red scale bar of 2 mm.

**Figure 3 polymers-14-01022-f003:**
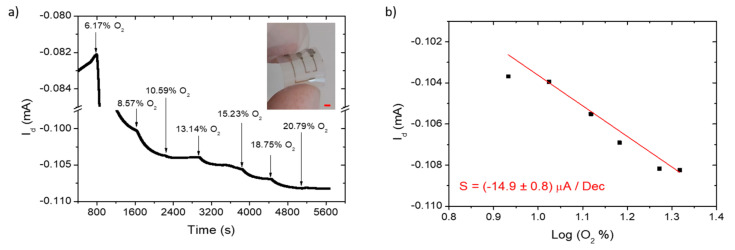
H-Thin OECT on PEN substrates for oxygen sensing: I_d_(t) measurement (**a**); sensor calibration curve (**b**); for oxygen addition (black arrows) in the sealed cell, starting from a nitrogen-saturated environment (0% of O_2_). The device picture is reported in the inset of (**a**); with a red scale bar of 2 mm.

**Figure 4 polymers-14-01022-f004:**
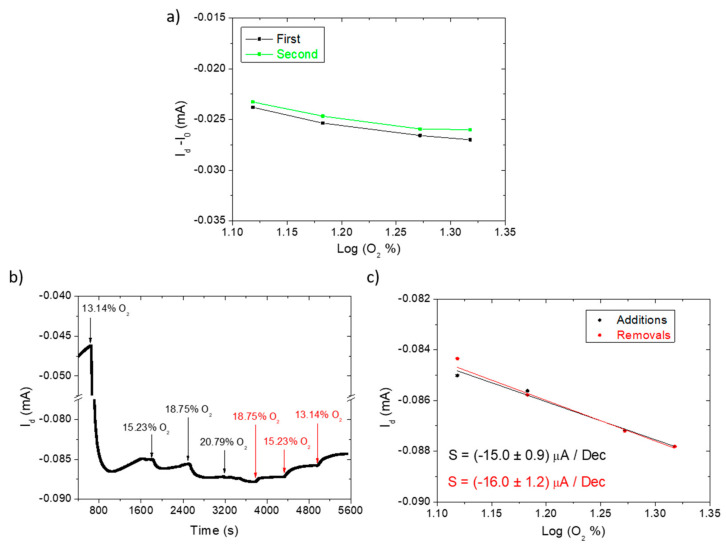
(**a**) Repeatability test comparing two subsequent measurements of an oxygen sensing OECT, in the range 13–21%. I_0_ represents the output current when no oxygen is present in the electrochemical cell and was subtracted to consider the slight baseline shift of the device; (**b**) I_d_(t) for oxygen additions and removals, marked by the black and red arrows, respectively; (**c**) corresponding sensor calibration curve.

**Figure 5 polymers-14-01022-f005:**
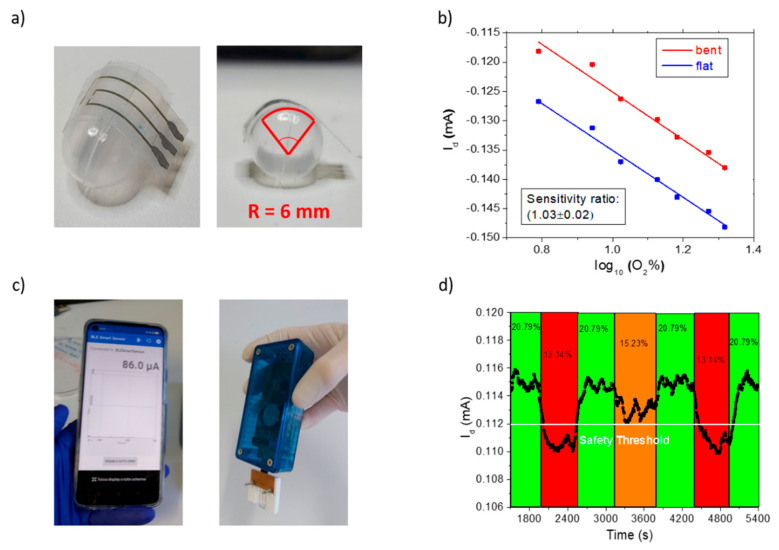
Bending test: (**a**) OECT patterned on PEN foil bent with a radius of 6 mm to mimic a real-life application; (**b**) OECT sensing the oxygen additions in the complete range from 6% to 21% in the flat (blue) and bent (red) configuration: (**c**) interfacing the sensor with a portable, handheld, battery-powered electronic readout from Elements srl company, wirelessly transmitting the current signal to a smartphone application; (**d**) warning test on a bent sample connected to the Elements system, showing the current response upon oxygen level variations (oxygen percentages are reported in black in the corresponding colored interval time). A safety threshold has been highlighted in white at 15% O_2_ to ensure operator’s protection before oxygen depletion could cause breathing fatigue or fainting.

**Table 1 polymers-14-01022-t001:** OECT parameters and currents extracted from the pulsed and transfer measurements for the three selected electrolytes (mean ± standard deviation).

	PBS	H-Thick	H-Thin
τ [ms]	1.005 ± 0.006	1.12 ± 0.03	198.21 ± 0.13
$I_{g}\;\left(=0\;V\right)\;$[µA]	−0.31 ± 0.06	−0.27 ± 0.17	−0.0258 ± 0.0012
$I_{g}\;\left(=0.3\;V\right)\;$[µA]	−0.20 ± 0.07	−0.14 ± 0.10	−0.0165 ± 0.0013
$I_{d}\;\left(=0\;V\right)\;$[µA]	−46.69 ± 0.08	−58.27 ± 0.10	−30.818 ± 0.007
$I_{d}\;\left(=0.3\;V\right)\;$[µA]	−25.34 ± 0.07	−41.90 ± 0.10	−18.297 ± 0.004
Gain = $I_{d}$/$I_{g}$ (0 V)	150	220	1195
Gain = $I_{d}$/$I_{g}$ (0.3 V)	130	300	1110
Gm peak [µS]	207	174	241

**Table 2 polymers-14-01022-t002:** Extracted sensitivities of three measured H-Thin OECT on glass substrates.

Sample	Sensitivity [µA/dec]	Error [µA/dec]
1	−13.6	1.3
2	−12.5	1.2
3	−11.2	0.9

**Table 3 polymers-14-01022-t003:** Extracted sensitivities of three measured H-Thin OECT on PEN substrates.

Sample	Sensitivity [µA/dec]	Error [µA/dec]
1	−15.0	0.9
2	−14.9	0.8
3	−16.1	0.7

## Data Availability

Data are available from the corresponding author on reasonable request.

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
