# Peer review of "Oxygen Gas Sensing Using a Hydrogel-Based Organic Electrochemical Transistor for Work Safety Applications"

_polymers, 2022, doi:10.3390/polym14051022_

Round 1

Reviewer 1 Report

The manuscript presents a detailed and extensive work on an oxygen sensor based on OECTs. The authors present a comprehensive work on an innovative sensor, based on previous work from the group that uses PEDOT-PSS as the active material. The sensor tested can be made on flexible substrate and has a good sensitivity, it is reproducible and measurements can be repeated. The presentation and English are very good. 

Author Response

We thank the Reviewer for his/her comment on our work.

Reviewer 2 Report

The authors present a study on hydrogel-PEDOT:PSS electrochemical transistor as a promising oxygen sensor for work safety applications.

The manuscript is concise, well-written, the methodology is adequate and the results are interesting.

However, in my opinion, the manuscript needs further revision before it can be accepted for publication in Polymers.

In the following lines, I list some queries:

Q1) Page 2, line 75-77.

Quote "...The energetically preferable reaction is the following:

4?????0+ ?2+ 4?+→4?????++2?2?  (1)

which causes PEDOT:PSS switching from the neutral to the oxidized state..."

This statement needs revision.

PEDOT is already partially oxidized in PEDOT:PSS due to PSS doping. Is the equation valid for partial de-doped PEDOT:PSS due to prior cation injection? What is the role of protons in the doping of PEDOT:PSS?

According to Ref29 in this manuscript [Singh et al, Journal of Physical Chemistry C, 121(22), 12270–12277] protons can bond to O2 to form the OOH+ specie but also, isolated protons have shown doping ability in PEDOT according to He et al [Polymers 2018, 10, 1065; doi:10.3390/polym10101065].

Q2) Page 3, line 123-132.

The difference between PBS and the thick hydrogel is not obvious, please clarify. Also, be specific about which ions are working as electrolytes, is it potassium, sodium, or other?

Q3) Page 4, line 172.

Please define briefly Ag/Ach parameter.

Q4) Page 8, line 292-295.

It is mentioned that the current difference between flat and bent configurations is observed after repeated measurements. However, can authors provide results on the current versus time while performing a flat-bent-flat-bent sequence to evaluate a more realistic dynamic scenario?

Author Response

We thank the Reviewer #2 for his/her comments, please find the attachment with our answers.

Reviewer 3 Report

Dear Editor, in the present work authors describe a hydrogel-based Organic Electrochemical Transistors (OECTs) made with the conducting polymer poly(3,4-ethylenedioxythiophene):poly(styrene sulfonate) (PEDOT:PSS) as wearable and real-time oxygen gas sensors. The paper is well described and provides new and interesting data. For this reason I propose to accept it for publication in following you can find some minor importance comments.

What is the advantage of the proposed OECTs, compared with the already existed on the market or described in other published works. This is not clear in introduction and in the aim of this work.

There are some mistakes in the text that should be corrected. For example, O2 should be corrected to O2.

Why these polymeric materials and these rations have been chosen? This this also not clear in the paper.

Author Response

We thank the Reviewer #3 for his/her comments, please find attached our answers.

Round 2

Reviewer 2 Report

Authors have addressed my queries and provided a revised manuscript which, in my opinion, is now ready to be accepted for publication in Polymers.